# Multiple Signatures of the JC Polyomavirus in Paired Normal and Altered Colorectal Mucosa Indicate a Link with Human Colorectal Cancer, but Not with Cancer Progression

**DOI:** 10.3390/ijms20235965

**Published:** 2019-11-27

**Authors:** Elena Uleri, Claudia Piu, Maurizio Caocci, Gabriele Ibba, Francesca Sanges, Giovanna Pira, Luciano Murgia, Michele Barmina, Simone Giannecchini, Alberto Porcu, Caterina Serra, Antonio M Scanu, Maria R De Miglio, Antonina Dolei

**Affiliations:** 1Department of Biomedical Sciences, University of Sassari, viale san Pietro 43B, 07100 Sassari, Italy; elenauleri@tiscali.it (E.U.); claudia-piu@tiscali.it (C.P.); maurizio.caocci@yahoo.it (M.C.); G.ibba@gmx.com (G.I.); f.sanges@yahoo.it (F.S.); gpira@uniss.it (G.P.); cserra@uniss.it (C.S.); 2Department of Experimental Surgery and Medical Sciences, University of Sassari, viable san Pietro 8, 07100 Sassari, Italy; lmurgia@uniss.it (L.M.); michelebarmina@gmail.com (M.B.); alberto@uniss.it (A.P.); scanu@uniss.it (A.M.S.); demiglio@uniss.it (M.R.D.M.); 3Department of Experimental and Clinical Medicine, University of Florence, viable Morgagni 50, 50134 Florence, Italy; simone.giannecchini@unifi.it

**Keywords:** colorectal cancer, adenomatous polyps, JC polyomavirus, T-antigen, *NCCR* non-coding control region, oncogenesis

## Abstract

The JC polyomavirus (JCV) has been repeatedly but discordantly detected in healthy colonic mucosa, adenomatous polyps, and colorectal cancer (CRC), and proposed to contribute to oncogenesis. The controversies may derive from differences in JCV targets, patient’s cohorts, and methods. Studies of simultaneous detection, quantification, and characterization of JCV presence/expression in paired samples of normal/altered tissues of the same patient are lacking. Therefore, we simultaneously quantified JCV presence (DNA) and expression (mRNA and protein) of T-antigen (*T-Ag*), Viral Protein 1 (*Vp1*), and *miR-J1-5p* in paired normal/altered tissues of CRC or polyps, and from controls. JCV signatures were found in most samples. They increased in patients, but were higher in normal mucosa than in corresponding polyp or CRC lesions. JCV non-coding control region *(NCCR)* DNA rearrangements increased in CRC patients, also in normal mucosa, thus before the onset of the lesion. A new ∆98bp *NCCR* DNA rearrangement was detected. *T-Ag* levels were higher in normal mucosa than in adenoma and adenocarcinoma lesions, but decreased to levels of controls in established CRC lesions. In CRC, *miR-J1-5p* expression decreased with CRC progression. *Vp1* expression was not detected. The data indicate a JCV link with the disease, but possible JCV contributes to oncogenesis should occur at pre-polyp stages.

## 1. Introduction

Colorectal cancer (CRC) is the third most common malignancy worldwide, and the second leading cause of cancer deaths [1,2], with a rising incidence at younger ages and some heterogeneity between countries [3]. Both genetic and environmental factors are involved in CRC etiology [3,4], including smoking, alcohol intake, increased body weight, and type-2 diabetes. Worldwide, several independent studies have reported the presence of genomic sequences of the JC polyomavirus (JCV) and expression of its potentially oncogenic T-antigen (T-Ag) protein in tissues from adenomatous polyps and colorectal adenocarcinomas, but also in normal tissues and in adjacent noncancerous tissue from the gastrointestinal tract [5]. At least eight independent groups [5,6,7,8,9,10,11,12,13] detected JCV signature in CRC (in 28–90% of the samples) [5,7,10,14], however, negative results were also found [15,16].

Some human polyomaviruses have been associated with cancer. The oncogenic role of the Merkel cell polyomavirus in skin Merkel cell carcinoma has been ascertained, and it is suspected for the BK polyomavirus in bladder carcinoma [17,18], with a possible “hit and run” mechanism [19].

JCV is an opportunistic pathogen [17,20,21,22], infecting 70–90% of humans, usually in early childhood, and it persists generally as an innocuous bystander. Primary infection is subclinical; kidneys, B-lymphocytes, and astrocytes may serve as sites for latency. Occasionally, JCV is released in the urine of healthy people or under mild-to-severe immune alterations, as in pregnancy and transplants. Rarely, in severe pathological or therapeutic immunosuppression, JCV infects brain oligodendrocytes lytically, causing the fatal progressive multifocal leukoencephalopathy (PML).

The JCV genome has a regulatory non-coding control region (*NCCR*), and early and late coding regions, leading, respectively, to the large-T and small-t antigens (T-Ag and t-Ag), and to the multifunctional agnoprotein (agno) and Viral Protein 1 (Vp1), Viral Protein 2 (Vp2), and Viral Protein 3 (Vp3) capsid proteins [23]. Although JCV replication is restricted to humans, its inoculation in some mammalian species is tumorigenic [22], mainly due to the early antigens. Similarly to simian vacuolating virus 40 (SV40) T-Ag, JCV T-Ag can bind and break DNA, has helicase and ATPase activities, and dysregulates the cell cycle by sequestering the Rb and p53 tumor suppressor proteins [22]. The T-Ag translocates IRS-1 to the nucleus, where it interferes with DNA repair fidelity, leading to the accumulation of mutations [24]. Moreover, in cultured colonic cells, JCV increases the AKT and MAPK activities, two pathways involved in tumorigenesis. JCV has been associated with chromosomal instability (CIN) and aneuploidy in tumors. Khalili and colleagues showed that T-Ag interacts with β-catenin, suggesting that T-Ag stabilizes β-catenin before loss of tumor suppressor genes, allowing CIN to emerge [22]. Ricciardiello’s group showed that in the diploid RKO colon carcinoma cell line, JCV-transient transfection induces chromosomal breakage, di-centric chromosomes, and increased ploidy, with a “hit and run” mechanism that involves an early interaction with β-catenin and p53 [25]. These features make JCV unique in its ability to simultaneously disrupt chromosomal integrity and abolish cell cycle checkpoints.

The JCV DNA was detected in 90% of colorectal tumor lesions with respect to 48% of normal surrounding mucosa, suggesting a selection for virus-containing cells at some early stage in tumor initiation or progression [26]. Recently, JCV DNA was found in 46% of the adenocarcinomas, but in none of the normal biopsies of either CRC or control patients; moreover, its presence was correlated with tumor location and grade [27]. In another study from the same country, JCV DNA was detected in 58% of CRC and in 15% of paired non tumor samples, and JCV presence was significantly correlated with tumor differentiation, as well as accumulation of β-catenin and p53 [14]. The JCV DNA was detected also in 20–89% of normal mucosa samples from controls and patients [28,29]. However, JCV DNA copy numbers were higher in tumor mucosa, and JCV proteins were detected only in neoplastic tissues. Several groups also found JCV in brain tumors, as glioblastoma and medulloblastoma [8]; JCV was actively responsible for cell transformation in cell lines obtained from tumors induced by JCV intracranial inoculation in hamsters and transgenic mice [30]. JCV genome and T-Ag have been found in colorectal carcinoma, but also in benign pre-neoplastic polyps, suggesting that JCV may play a role in the early stages of the neoplastic process [28,29]. However, other studies did not detect viral genomes, neither in hyperplastic polyps/adenoma and adenocarcinoma, nor in normal adjacent and non-adjacent mucosa [15,31].

In metastatic primary CRC, T-Ag DNA was found in corresponding liver metastasis, and JCV was associated with a metastatic phenotype [32]. These data are supported by an in vitro study, showing increased cell migration when colonic cell lines were transfected with a T-Ag expression plasmid [33]. In this view, one can argue that the presence of viral DNA in colorectal lesions is also associated with its previous presence in normal colonic mucosa. A study reported anti-T-Ag immunoreactivity in 46% of CRC without JCV DNA, concluding that the antibody cross-reacted with an undefined protein, whose expression was associated with chromosomal instability, lymph node metastasis, and a more advanced tumor stage [34]. Another qualitative study showed JCV DNA in 44% of CRC in the tumor mass (and 35% also in surrounding tissue), but also in infiltrating lymphocytes [12]. Stromal JCV T-Ag DNA signals in CRC were reported in association with short survival and bad prognosis [35]. Another group detected T-Ag-specific Th-1 cells in all patients with colorectal polyps or cancer, but the Th-1 response in cancer patients was much lower [36].

Of relevance is the discovery of a JCV variant only in colon cancer tissues. JCV exists in two forms: the non-pathogenic archetype, persisting in the kidney, and the PML-type Mad-1 or Mad-4 variants, which were detected in all the aforementioned tumor samples. The difference lies within the *NCCR* region—the PML-type *NCCR* contains a duplication of a 98bp sequence, absent in the archetype, which functions as an enhancer for transcription. In a significant number of CRC samples, but not in the adjacent non-neoplastic tissues, the Mad-1 strain of JCV showed a variant with a single 98bp sequence [13].

The JCV-specific *miR-J1-5p* microRNA (miRNA), which downregulates T-Ag expression [37], was evaluated in healthy and CRC specimens, and detected in all paired CRC and normal tissues, with a prevalence of lower *miR-J1-5p* levels in tumor tissues [38].

CRC is a global emergency with respect to incidence and mortality; therefore, the possibility of a carcinogenic role of JCV deserves to be clarified.

The published studies on JCV in CRC are discordant, and only few of them analyzed T-Ag expression in paired samples. The controversies may derive, at least in part, from evaluation of different targets, different design, and/or methods. Studies of simultaneous detection, quantification, and characterization of JCV presence/expression in samples of normal/altered tissues of the same patient are lacking. To give insights on JCV role in CRC, we simultaneously analyzed JCV DNA presence and genotype, and the expression of T-Ag, *Vp1*, and *miR-J1-5p*, in paired normal/altered tissues from patients with CRC and polyps, as well as from controls. Our study is the first to report a complete analysis of JCV signatures in samples of CRC and adenomatous polyp pairs and in non-tumor controls, measuring (i) *NCCR* and T-Ag DNA, (ii) the expression of T-Ag (mRNA and protein) and *Vp1*, and (iii) *miR-J1-5p*.

## 2. Results

### 2.1. Detection and Analysis of JCV DNA

We evaluated the T-Ag gene, as it codes for a protein with oncogenic properties, and the *NCCR*, as it contains the domain prone to variability from which the transcription efficiency depends. We had enough material to examine the DNA of 30 CRC paired samples, 9 polyps, and 9 controls (Table 1). As described in the Methods section, the real-time PCR for T-Ag DNA detection was performed on 100 nanograms of each sample, with specific primers and probe, followed by normalization to the glyceraldheyde-3-phosphate dehydrogenase (GAPDH) housekeeping gene. The *NCCR* sequences were amplified by nested PCR on 200 nanograms of each sample. The amplicons were isolated on agarose gel, purified, and cloned into a cloning vector. Ligation products were used to transform competent cells, and six colonies from each sample were sequenced with the Sanger method.

The T-Ag DNA was detected in 93% of CRC cases (both in tumor and in normal mucosa), in 100% of polyps, and 89% of controls (Table 1). As shown on Figure 1, the mean T-Ag levels were 2.4-fold more abundant in normal mucosa than in tumor mucosa (*p* = 0.02, Figure 1A). With respect to individual CRC paired samples, 22 CRC pairs showed reduced levels in the lesion compared to the normal counterpart, whereas eight CRC pairs showed the opposite (Figure 1B, statistical significance: *p* = 0.009 by Wilcoxon test for paired samples). No statistically significant association between T-Ag DNA behavior within CRC pairs and clinical stage was observed (not shown).

By *NCCR*-specific nested PCR, 45% of CRC cases were positive both in the tumor and in the normal part; 26% were completely negative and 30% were positive only in one part. Considering the double-positive CRC cases, and those positive only in one of the two parts, JCV *NCCR* was found in 74% of the CRCs tested. All the polyps and the controls tested contained the *NCCR* region (100%) (Table 1).

The *NCCR*-specific nested PCR produced two types of amplicons (Figure 2A), one of the size of the PML-type (353bp) and a smaller one (260bp). These amplicons were sequenced and compared to *NCCR* sequences of the NCBI database (accession numbers AB372038.1, NC_001699.1, AB220941.1, AF300953.1), as reported in Figure 2B. All the sequenced samples (22 CRC cases, 9 polyps, and 8 controls) contained a rearranged *NCCR* of the size and organization typical of the PML-type, with the classical 98bp duplication.

The smaller band was a rearrangement of the *NCCR* Mad-1PML-type, with a 98bp deletion, which we named ∆98-type, according to *NCCR* nomenclature [40]. We observed two types of ∆98bp rearrangements: a ∆98a variant (with deletion of the second 98bp domain), already described in CRC tumor mucosa [13], and a new ∆98b rearrangement (with deletion of the first 98bp domain, and conjunction between the A region of the first tandem repeat with the C region of the second one). Figure 2B reports a schematic representation of the PML-type *NCCR* organization and the alignment to JCV strains from the NCBI database, showing representative sample sequences from controls, polyps, and CRC (four patients each). Point mutations were also detected, mainly in the first A region (where the TATA box is) and in the C region.

We detected the ∆98bp variants in both healthy and tumor tissues as follows: 74% of tumor mucosa, 67% of adjacent mucosa, and 46% in both. Two CRC cases had only the ∆98bp variant, one in the adjacent part, the other in the tumor part. Regarding the polyps, 33% of them also showed the ∆98bp variant, in addition to the PML-type *NCCR*, whereas it was present in 44% of controls. Figure 2C reports the percent distribution of the *NCCR* PML-type and ∆98bp variants in the groups; the latter differed significantly between lesion tissues of CRC and polyps, (two tailed Fisher’s exact test, *p* = 0.045).

### 2.2. JCV Expression

The levels of *T-Ag* and *Vp1* mRNAs were evaluated by RT-real time PCR, as described [39], in 41 CRC and 7 polyp normal/lesion mucosa pairs. The mRNAs were reverse-transcribed and amplified as described in the Methods section. When samples were exposed to real time PCR with *Vp1*-specific primers, no expression of *Vp1* was detected, as reported in Table 1, indicating the absence of viral replication.

Data of *T-Ag* transcripts in CRC pairs are reported in Figure 3A, and show that *T-Ag* mRNA levels were higher in normal mucosa than in the corresponding tumor part (*p* = 0.009, by Wilcoxon test for paired samples). The study of paired polyps and corresponding normal mucosa (left graph of Figure 3A) also revealed that in the normal tissue of patients with polyps, the *T-Ag* was transcribed more than in the corresponding lesion (*t*-test *p* = 0.015).

To detect the T-Ag protein, which is the functional stage, we performed western blotting analysis of those samples that yielded enough for this assay also (16 CRC, 9 polyps, and 9 controls) by using a monoclonal antibody directed against the N-term region of the T-Ag. Representative blots are shown in Figure 3C, which reports the 78 kDa band of T-Ag and the 42 kDa band of the β-actin housekeeping control. To quantify the T-Ag protein in the bands, and to compare T-Ag levels of CRC, polyps, and controls, a band intensity analysis was performed, as reported in the Methods section.

The data of individual CRC pairs are shown in Figure 3B—as expected from the *T-Ag* mRNA trends, the levels of the T-Ag protein were significantly higher in the normal mucosa than in the lesion (Wilcoxon matched paired test, *p* = 0.043). Figure 3D reports the normalized T-Ag protein levels of all the groups.

The comparison between the band intensities of controls and polyps showed that the T-Ag expression levels of polyps were 2.3-fold higher than those of the unpaired controls (mean AU (arbitrary units): 0.4 and 0.92, for controls and polyps, respectively, *t*-test *p* = 0.009), and more scattered. Interestingly, the T-Ag expression levels of normal mucosa of CRC pairs (mean AU: 0.87) were about the same detected in the polyps, whereas the CRC tumor part had a strong T-Ag decrease with respect to the normal part (mean AU: 0.48, *p* = 0.023), almost down to the levels of the controls.

### 2.3. miR-J1-5p Evaluation

The *miR-J1-5p* were analyzed on 39 CRC pairs (Figure 4). In Figure 4A, the data are expressed as tumor/normal 2^−∆*C*t^ ratio, showing that 44% of the CRC lesions expressed lower *miR-J1-5p* than the normal mucosa from the same patient, whereas 36% of the pairs showed the opposite; 20% of the pairs had minor variations. Some of the samples had enough material to be analyzed also for *T-Ag* expression by western blotting, and could allow the comparison of *miR-J1-5p* levels to those of the *T-Ag* protein, as an inverse relationship between *T-Ag* protein expression and *miR-J1-5p* occurs during JCV replication [37]. However, only 50% of CRC cases presented this inverse relationship (Figure 4B). When *miR-J1-5p* data were compared to *T-Ag* RNA, DNA, and clinical data, only the correlation with TNM (tumor-node-metastases) staging emerged—the *miR-J1-5p* tumor/normal ratio decreased with the CRC progression (Figure 4C).

## 3. Discussion

In search of a link between JCV and CRC oncogenesis, many independent studies evaluated JCV presence and/or expression or sero-reactivity in CRC and adenomatous polyps. Altogether, the studies found JCV signature in lesions, in adjacent normal tissues or in non-tumor colorectal mucosa, with discordant findings, attributable, at least in part, to evaluation of different targets, and by different methods; only few studies analyzed paired samples from the same patient [5,7,10].

We performed a multi-parametric analysis of JCV in paired samples of normal and altered mucosa from patients with CRC or polyps, compared also to controls. To our knowledge, it is the first study that simultaneously evaluates and quantifies JCV presence (DNA, the *T-Ag*, and *NCCR* regions) and expression (mRNA and proteins, *T-Ag*, and *Vp1*) in the same sample, and compares pairwise normal and altered tissues from the same individual, thus excluding bias due to variability between individuals, and to different targets and detection methods.

According to our data, *T-Ag* DNA detection in colorectal mucosa is a common finding, occurring in 89% of controls, 100% of polyps, and 93% of CRC cases (Figure 1A). The pairwise analysis gave a new finding—the positivity was in both normal mucosa and lesion, and the few negative pairs were so in both parts. Therefore, it is likely that the local infection occurred before CRC onset. The controls had the lowest T-Ag DNA levels, as one would expect, if JCV has some link to the disease. The *T-Ag* DNA levels of polyps were 2.6-fold higher than controls (*p* = 0.01). However, the highest *T-Ag* DNA levels were observed in the normal mucosa of CRC patients (4.5-fold more than controls, *p* = 0.001). In turn, the normal mucosa of CRC pairs had 2.4-fold more *T-Ag* DNA than the lesion (*p* = 0.009), even though some CRC pairs showed a slight increase in the tumor part. Because the mean *T-Ag* DNA levels of polyps were similar to those of CRC lesions, one can argue that, after an initial increase, JCV was reduced in tumor tissues. Given the statistically significant differences among the low JCV load of controls and the CRC normal tissues, as well as among polyps and controls, it is possible that a high JCV load could predispose to cell transformation. We are aware that our polyps and controls derive from different individuals and cannot be representative of the adenoma tumor progression.

To our knowledge, this is the first report of higher JCV DNA levels in normal compared with corresponding CRC mucosa, and the data are confirmed by those of mRNA and protein expression (see below). By real-time PCR, Newcomb et al. [15] did not find any JCV positivity in matched CRC tumors. By nested PCR of serial dilutions from paired samples, Laghi et al. [28] found higher levels in CRC tissues, with 26–89% positivity depending on the method used. By simple or nested PCR, Toumi et al. [27] found JCV DNA exclusively in tumors (46%), but not in normal mucosa, and hypothesized differences among patients from different countries. By simple PCR, Casini et al. found 89% JCV/BKV positivity in Italian CRC patients, mostly in both tumor and normal mucosa [12], whereas no JCV DNA was found in Italian CRC patients by Campello et al. [31] by real-time quantitative PCR. According to us, Italian individuals behave as other cohorts.

In a report on gastric cancer, JCV *T-Ag* DNA of both cancer tissues and paired adjacent normal parts was 10-fold higher than in normal gastric patients [41].

In our study, the positivity for *NCCR* and *T-Ag* DNAs was the same in controls and polyps, whereas, in overall CRC, it was 74% for *NCCR* and 93% for *T-Ag*. This discordance could be intrinsic to the virus and/or the disease stage, or might be due to the different methods used.

Two types of *NCCR* amplicons were observed (Figure 2A): one of the size of the PML-type (369bp) and a smaller one (270bp). By sequencing, the higher band was found to be the expected PML-type Mad-1 prototype. The lower band was a Mad-1 variant, with two types of rearrangements. The ∆98a rearrangement, with deletion of the second 98bp domain, has been observed in CRC tumor mucosa [13], in kidneys and brains of infected patients [42], and in bone marrow and brain tissue from a PML patient [43]; a construct with this rearrangement was shown to transform Rat-2 fibroblasts more efficiently than the Mad-1 prototype [44]. We found also a new ∆98b rearrangement (Figure 2B). Functionally, the two forms should behave similarly, as they share the same binding sites for transcription factors.

Although the Mad-1 prototype prevailed in controls and polyps, the rearranged forms were increased in CRC normal and lesion samples (polyps versus CRC lesions: *p* = 0.045, Figure 2C). DNA rearrangements are expected during oncogenesis. However, because *NCCR* rearrangements were increased also in CRC normal mucosa, the data suggest that the *NCCR* rearrangements *preceded* the CRC transformation and thus could be a feature of persons who develop CRC.

The analysis of the transcriptional activity of JCV genes did not detect expression of *Vp1* in CRC pairs, indicating the lack of JCV replication in colonic cells from these patients.

*T-Ag*, instead, was actively expressed both as mRNA and protein (Figure 3). The present study is the first quantifying *T-Ag* expression at both levels, and pairwise. In both polyp and CRC pairs, the mean *T-Ag* mRNA levels were 2.4-fold higher in the normal mucosa than in the lesion. In the 16 CRC pairs, which yielded enough in amount to allow protein analysis, the levels of the T-Ag protein were consistent with those of transcripts. Importantly, the levels of the T-Ag protein, which is the most relevant JCV protein for a contribution to oncogenesis, varied with disease status in a statistically significant manner. The lowest levels were those of the controls, as one would expect. In polyps, the values of T-Ag protein were more than twice those of controls. Interestingly, the *T-Ag* expression levels of normal mucosa of CRC pairs were about the same detected in the polyps, whereas the CRC tumor part had a strong *T-Ag* decrease with respect to the non-lesion part and to polyps, almost down to the levels of the controls. There have been only four studies that have analyzed T-Ag protein presence in colorectal mucosa, all by immunostaining of formalin-fixed, paraffin-embedded samples from lesions, without the corresponding normal mucosa. Three of them analyzed polyps, and found 0% [35], 5% [45], and 16% [29] positivity. Two studies analyzed CRC samples, finding 0% [35] and 65% [46] positivity.

Our data indicate that *T-Ag* expression is measurable in almost all samples, with wide variations among individuals (~4 logs at the mRNA level, and ~2.5 logs at the protein level). The high sensitivity was primarily due to the fact that freshly excided tissues were analyzed, and because, to detect the T-Ag protein by western blot, proteins from many cells were collected, thus amplifying the signal. By immunostaining, instead, only cells expressing high amounts of protein can be detected.

The T-Ag protein was shown to induce chromosomal instability in colonic cells, with early inactivation of p53 and Rb tumor suppressors, dysregulation of signaling pathways, and interference with DNA repair [22,25,47]. Therefore, our findings are in line with the possibility of a “hit and run” mechanism of JCV contribution to cell growth dysregulation and chromosomal instability. According to this model, a pre-existing JCV infection is associated with the early stages of cell growth alterations, but it is no more needed for the progression when additional mutations render T-antigen expression dispensable.

JCV miRNAs have been proposed as a biomarker for JCV infection in the gastrointestinal tract [38]. In CRC pairs, we found expression of JCV-specific *miR-J1-5p* in normal and tumor mucosa of all samples tested (Figure 4), with both lower and higher expression in the tumor part, with respect to normal mucosa from the same patient, in line with previous data [38]. JCV *miR-J1-5p*, produced during late transcription, controls viral replication through *T-Ag* downregulation and targeting of host factors, helping JCV to escape the immune surveillance [37,48]. In our study, only 50% of CRC cases had the inverse relationship between *T-Ag* protein and *miR-J1-5p*, which occurs during JCV replication [37], indicating that in colonic cells and in a tumor environment, the expected *T-Ag* regulation by JCV miRNA does not occur. When *miR-J1-5p* data were stratified with respect to the TNM staging of the disease (Figure 4C), the tumor/normal ratio decreased with CRC progression, indicating that *miR-J1-5p* levels decrease more in tumor than in normal mucosa (at stage I, there were almost equal amounts in both, whereas at stage IV, the lesion had only 1/10 *miR-J1-5p* levels of normal mucosa).

The main conclusions of the present study are that (i) JCV infection of colorectal mucosa is much higher than expected. (ii) JCV is linked to colorectal disease, as viral presence (DNA) and expression (mRNA and protein) change with levels in normal adjacent mucosa higher than in the lesion of the same individual, concordantly and in a statistically significant manner, in controls and in patients with polyps or CRC. JCV *NCCR* rearrangements increase in CRC individuals, also in normal mucosa, thus before the onset of the neoplastic lesion. (iii) *T-Ag* presence and expression occur before the onset of the adenoma and adenocarcinoma lesions, as *T-Ag* levels are higher in the normal adjacent mucosa than in the lesion, but go down to the levels of controls in already established CRC lesions. (iv) The relative reduction of *miR-J1-5p* expression in CRC lesions is related to CRC progression.

## 4. Materials and Methods

### 4.1. Patients

Fresh samples from 41 CRC adenocarcinoma patients, 16 with adenomatous polyps, and 9 non-tumor controls (NTC) were obtained from the General Surgery Unit-2, Department of Clinical and Experimental Medicine, University of Sassari. The study was approved by the Bioethics Committee of the Azienda Sanitaria Locale Sassari, Italy (# 2032/CE, 13.05.2014), and all participants gave informed consent. Paired samples from the lesion and from adjacent normal mucosa (7–10 cm faraway, without any sign of alteration) were obtained from all the CRC patients, and from seven of the polyp cases. The tumors were classified according to the TNM (tumor-node-metastases) staging [49]. Demographic and clinical-pathological data are reported in Table 1. The samples were processed within 1–2 h from excision. Depending on the amounts available, each sample was divided in up to four parts for DNA, RNA, and protein extraction. To preserve RNAs and proteins, the corresponding aliquots were stored in RNA-Later (Thermo-Fisher Scientific, Inc., Waltham, MA, USA) at +4 °C overnight. Then, the RNA-Later was removed, and the samples were transferred at −80 °C. The samples for DNA extraction were stored dry at −80 °C.

### 4.2. DNA Analysis

DNA extraction was carried out with DNeasy Blood and Tissue Kits, according to the manufacturer’s instructions (Qiagen, Hilden, Germany). Enough tissue amounts for this assay were available for 30 CRC pairs, 9 polyps, and 9 controls. The real-time PCR for *T-Ag* DNA detection was performed on 100 ng of each sample, as published [50]. Two-hundred ng of DNA were used to detect JCV *NCCR* by nested PCR, as published [51]. Five microliters of the first PCR product served as template for nested PCR. The amplicons were analyzed by electrophoresis on 1% agarose gel. The 353bp band and the lower band were excised from the gel; the DNA was purified by PureLink Quick Gel Extraction and PCR Purification Combo kit (Thermo-Fisher Scientific, Inc., Waltham, MA, USA) and cloned using TA cloning kit for sequencing (Life Technologies, Eugene, OR, USA) [42]. Ligation products were used to transform DH5α *Escherichia coli* competent cells. DNA was extracted from colonies by miniprep kit (Qiagen, Hilden, Germany). From each sample, six colonies were sequenced with the Sanger method [52].

### 4.3. RNA Analysis

After mechanical dissociation, normal/lesional pairs from 41 CRC and 7 polyps were processed with miRNeasy Mini Kit, according to manufacturer’s instructions (Qiagen, Hilden, Germany). The mRNAs were enriched, using the Dynabeads mRNA kit (Dynal Biotech, Oslo, Norway). For each mRNA sample, 1 μg was reverse-transcribed, as published [53]. Real-time PCR with specific primers and probes for *T-Ag* [50] and *Vp1* were used [54]. To verify the proper mRNA extraction and reverse-transcription outcomes, validated internal controls were used [53]. As positive JCV control, DNA from transiently transfected SW480 cells was used [39]. For each sample, the Ct (cycle threshold) value of the gene of interest was normalized to that of the glyceraldheyde-3-phosphate dehydrogenase (GAPDH) housekeeping gene; the data were expressed according to the 2^−Δ*C*t^ method [55].

The expression of *miR-J1-5p* was measured by the specific JCV miRNA-J1-5p quantitative stem-loop RT-PCR MiRNA assay (Assay ID 007464_mat, Life Technologies, Foster City, CA, USA), according to the manufacturer’s protocol. The reaction was performed in triplicate by using 10 ng of each RNA, and included negative controls (no template) and synthetized oligonucleotides as standards. The specificity and reproducibility of the assay has been published [37].

### 4.4. Protein Analysis

Enough tissue for this study was available for 16 CRC pairs, 9 polyps, and 9 controls. Whole cell extracts were obtained after lysis with Ripa Buffer (Thermo-Fisher Scientific, Inc., Waltham, MA, USA) and mammalian protease inhibitor cocktail (Roche, Basel, Switzerland), as described [53]. Western blot was performed as published [53], with anti-SV40 T-Antigen PAb416 monoclonal antibody (Merck KGaA, Darmstadt, Germania), diluted 1:200. As loading control, the β-actin protein was evaluated using mouse anti-β-actin monoclonal antibody (Sigma-Aldrich, Buchs, Switzerland), diluted 1:5000. Secondary antibody was horseradish peroxidase-conjugated IgG antibody, diluted 1:15,000 (Thermo-Fisher Scientific, Inc., Waltham, MA, USA). The membranes were developed with a chemoluminescent substrate (Supersignal West Femto Maximum Sensitivity Substrate; Thermo-Fisher Scientific, Inc., Waltham, MA, USA), exposed to Molecular Imager VersaDoc 3000 (BioRAD, Hercules, CA, USA), and acquired by QuantityOne Software (BioRAD, Hercules, CA, USA), as published [39]. The band intensities were evaluated through the ImageJ 1.51.S software [56].

### 4.5. Statistics

Descriptive analysis included the computation of means, standard deviations, and *t*-test for independent samples or Wilcoxon signed-rank test to compare related samples. Linear regression tests (Pearson’s test) were performed to quantify the strength of relationship between targets and variables.

## Figures and Tables

**Figure 1 ijms-20-05965-f001:**
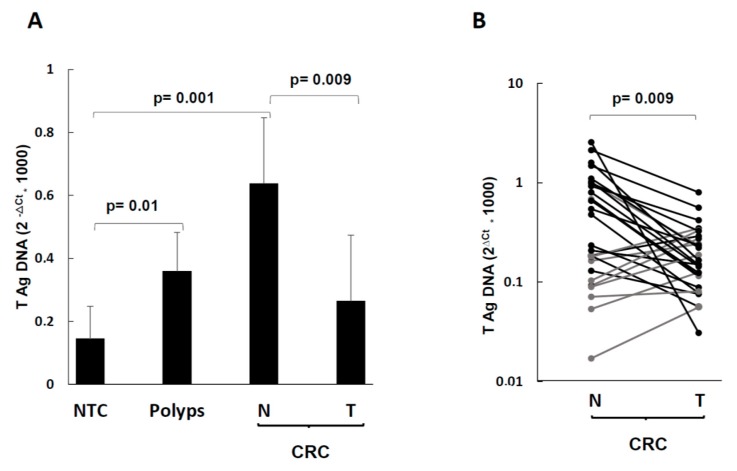
Levels of T-antigen (*T-Ag*) DNA in NTC, polyps, and CRC tumor and normal surrounding tissues. (**A**) Mean values of JCV *T-Ag* DNA copies in NTC, polyps, and CRC tumor (T) and normal (N) surrounding tissues; (**B**) individual values of JCV *T-Ag* DNA copies of paired tumor (T) and normal (N) surrounding tissues in CRC cases. Data are expressed according to the 2^−∆*C*t^ method [39]. The statistical significance was determined by the two-tailed Student’s *t*-test.

**Figure 2 ijms-20-05965-f002:**
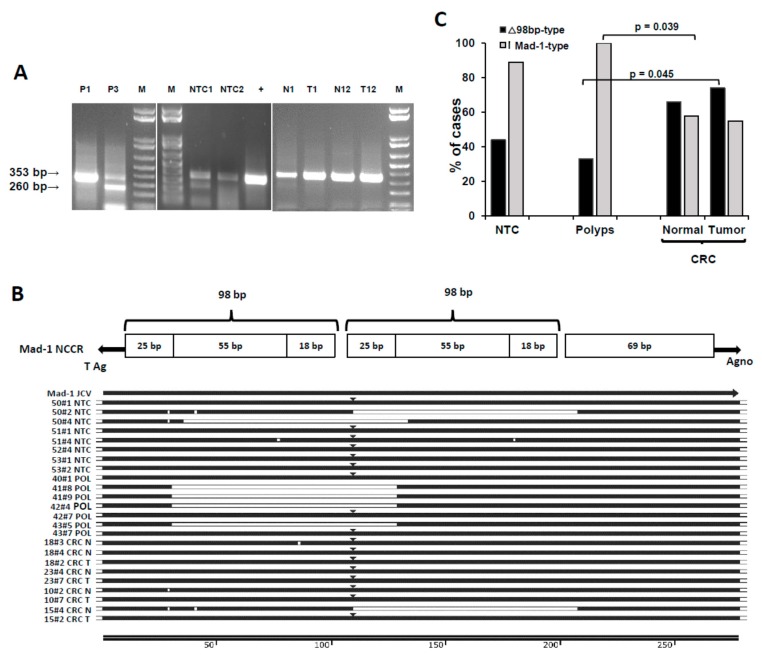
Presence and characterization of the JC polyomavirus (JCV) *NCCR* DNA in NTC, polyps, and CRC pairs. (**A**) Migration on agarose gels of representative *NCCR* amplicons, obtained by nested PCR, as specified in the Methods section. M: marker; +: positive control; NTC: non tumor control; P: polyp; N: normal adjacent mucosa; T: tumor mucosa. (**B**) *NCCR* multiple alignment of sample sequences to JCV strains from the NCBI database. At the top, a schematic representation of the PML-type *NCCR* organization is reported. (**C**) Distribution of the Mad-1 prototype and Δ98 rearranged *NCCR* forms in NTC, polyps, and CRC, expressed as percentage of cases. See text for details.

**Figure 3 ijms-20-05965-f003:**
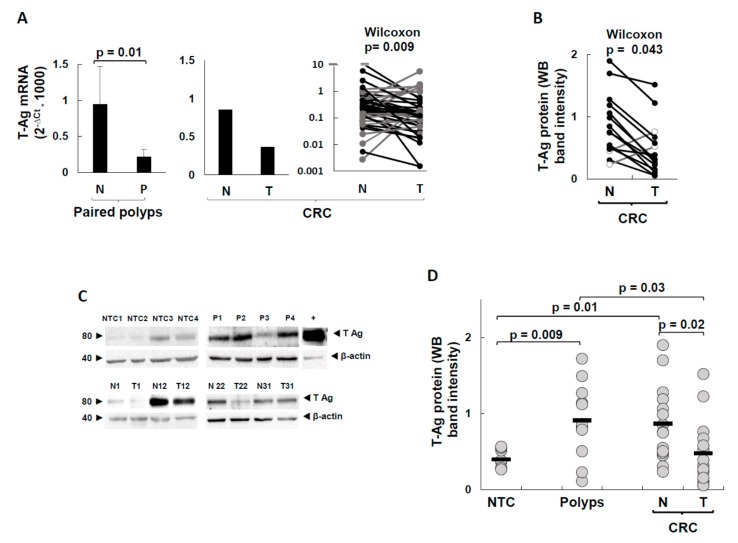
*T-Ag* expression in NTC, polyps, and CRC pairs. (**A**) Levels of *T-Ag* transcripts in paired adenomatous polyps and CRC cases. (**B**) Levels of T-Ag protein in CRC for each patient. The gray lines indicate the two samples with higher levels in the lesion than in normal tissue. (**C**) Western blotting of representative samples from non-tumor controls (NTC), polyps (P), and CRC normal (N) and tumor (T) tissues. (**D**) Arbitrary quantification of the intensity of relevant bands in western blotting for each sample of NTC, polyps, and CRC. The levels of *T-Ag* transcripts were expressed according to the 2^−∆*C*t^ method [39]. The statistical significance was determined by the two-tailed Student’s *t*-test in (**A**) and (**D**), and by the Wilcoxon signed rank test for paired samples in (**B**).

**Figure 4 ijms-20-05965-f004:**
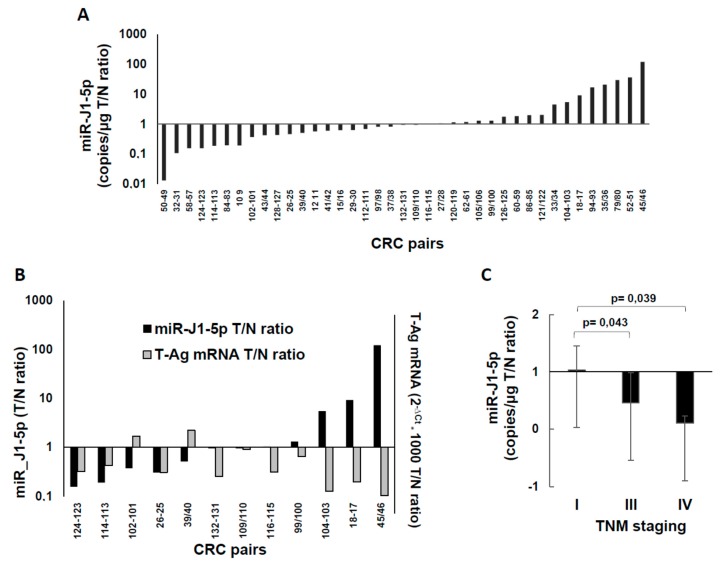
Levels of *miR-J1-5p* microRNA (miRNA) in CRC pairs. (**A**) *miR-J1-5p* values, obtained by a quantitative stem-loop RT-PCR miRNA assay, and expressed as CRC tumor/normal ratio. The values are sorted in ascending order. (**B**) Relationship among JCV *miR-J1-5p* miRNA and *T-Ag* expression; the relative miRNA expression in tumor and normal tissues (T/N ratio) is plotted against the corresponding *T-Ag* relative expression, expressed as T/N ratio. (**C**) *miR-J1-5p* CRC tumor/normal ratios stratified according to CRC TNM staging. The statistical significance was determined by two-tailed Student’s *t*-test.

**Table 1 ijms-20-05965-t001:** Clinical and Demographic Features of the Patients, and Summary of Evaluations of the JC Polyomavirus (JCV) DNA, RNA, and Proteins in Mucosal Samples from Non-Tumor Controls (NTC), Polyps, and Colorectal Cancer (CRC) Tissues.

Sample Types	NTC	Polyps	Paired Polyps and Adjacent Mucosas	CRC	
Mean age ± SD	58 ± 15	64 ± 8	62 ± 7	70.3 ± 9.7	
*n* Females/Males	9 4/5	9 4/5	7 4/3	41 16/25	
Histotype, *n*	NA, 9	Low grade dysplasia, 9	Low grade dysplasia, 7	Adenocarcinoma, 41	
Stage, *n*	NA	NA	NA	I, 10; II, 7 III, 20; IV, 4	
Mad-1 type *NCCR* ^a^ *n* (%)	9 positive (100)	9 positive (100)	nd ^b^	N+/T+; 13 (44) N+/T-; 4 (13) N-/T+; 5 (17) N-/T-; 8 (26)	30 *
T-Ag DNA ^c^, *n* (%)	8 positive (89)	9 positive (100)	nd	28 positive (93) 2 negative (7)	30 *
T-Ag RNA ^d^, *n* (%)	nd	nd	7 positive (100)	38 positive (95) 3 negative (5)	41 *
*Vp1* RNA ^d^, *n* (%)	nd	nd	nd	41 negative (100)	41 *
T-Ag protein ^e^, *n* (%)	9 positive (100)	9 positive (100)	nd	15 positive (94) 1 negative (6)	16 *

T-Ag: T-antigen; Vp1: Viral Protein 1; N: normal mucosa; T: tumor mucosa; NA: not applicable; *n*: number of samples tested; SD: standard deviation. The diagnosis was based on TNM (tumor-node-metastases) staging (stages I, II, III, IV). Detection methods: ^a^ non-coding control region (*NCCR*)-specific nested PCR, followed by sequencing; ^b^ nd: not done, for sample paucity; ^c^ real time PCR; ^d^ real time-RT PCR; ^e^ western blotting; * tested colorectal cancer (CRC) samples (the number of tested samples for each assay was dependent on the amounts available for each sample).

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
