# Peer review of "Multiple Signatures of the JC Polyomavirus in Paired Normal and Altered Colorectal Mucosa Indicate a Link with Human Colorectal Cancer, but Not with Cancer Progression"

_ijms, 2019, doi:10.3390/ijms20235965_

Round 1
Reviewer 1 Report
Uleri et al. performed a comprehensive analysis of JCV polyomavirus in normal mucosa and adenomatous polyps as well as in colorectal cancer tissue and autologous non-tumor tissue. They evaluated viral DNA presence and load by real time PCR targeting T-Ag and NCCR regulatory region showing that 90%-100% of samples were positive for the virus. Accordingly, the expression of T-Ag was lower in colorectal cancer than paired normal tissues, also confirmed by western blot. Furthermore, they analyzed NCCR amplicons and identified a new D98 variant. A decreased expression of miR-J1-5p with colon cancer progression has also been demonstrated.
The study is interesting and provides useful information on the role of JCV in colorectal neoplasia. The experimental design is well-planned and conducted, the results fully support the authors’ conclusions.
Minor comments:
Results
Lines 120-124. The authors describe "detection" of T-Ag gene and NCCR DNA sequences, however in figure 1 they described real time qPCR results. Therefore, this section of the results needs to be better described to explain the type of experiment they performed
Lines 152-153. The description of sequencing results implies an end-point PCR. As stated above a clearer presentation of the experimental procedure is needed in the results given that the journal format requires methods at the end of the manuscript.
There is no description of Vp1 and agnoprotein mRNA analysis in the results.
Author Response
Response to the Comments of Reviewer 1:
We thank this Reviewer for his/her comments. Our reply to the minor comments is as follows:
Q1: Lines 120-124. The authors describe "detection" of T-Ag gene and NCCR DNA sequences, however in figure 1 they described real time qPCR results. Therefore, this section of the results needs to be better described to explain the type of experiment they performed
R.: OK, the experimental methods were described better in the results paragraph 2.1 (new lines 124-129 of the revised version). We evaluated the T-Ag gene, since it codes for a protein with oncogenic properties, and the NCCR, since it contains the domain prone to variability, on which the transcription efficiency depends. We had enough material to examine the DNA of 30 CRC paired samples, 9 polyps and 9 controls (Table 1).
Q2: Lines 152-153. The description of sequencing results implies an end-point PCR. As stated above a clearer presentation of the experimental procedure is needed in the results given that the journal format requires methods at the end of the manuscript.
R.: The description of the procedures used has been described in the Results (new lines 124-129 of the revised version), before specifying the results obtained with the two methods.
Q3: There is no description of Vp1 and agnoprotein mRNA analysis in the results.
R.: In the lines 183-184 of the previous version, paragraph 2.2 of the Results, we stated that the mRNA analysis of both transcripts gave negative results. We thought that it was unsuitable to insert a figure with only negative, flat, data. Thus, we put them only on the Table 1. This is better specified now, in the revised paper, on lines 189-191. Moreover, due to the criticisms of Reviewer 2, we decided to eliminate the (negative) data about agno expression.
Reviewer 2 Report
Manuscript by Uleri et al. (ijms-610739) described the studies on possible association of JC polyomavirus (JCPyV) with colorectal cancer (CRC). They took biopsy samples from normal and altered colorectal mucosa and systematically analyzed for a sign of JCPyV gene expression. They found that the level of T-antigen DNA and its expression was higher in samples from CRC patients than those from healthy donor. Interestingly the sign of JCPyV presence in the normal mucosa from the CRC patient was more prominent than that in the lesion suggesting a link between JCPyV infection and tumorigenesis. This reviewer finds this manuscript organized, easy to read, and providing sufficient evidence to support authors’ conclusions. With minor modifications listed below this manuscript affords publication in IJMS.
Specific points.
Table 1 is highly confusing. The authors should devise way for better presentation. Alignment error of the numbers is unacceptable. Misplaced emphasis of data must be corrected. The significance of the table lies on the presence of JCPyV detected by DNA, RNA, and protein in each sample. Simply showing (+) or (-) attached after the numbers was not good way to attract readers’ attention. Creating categories, positive/ negative, may improve the clarity.
The reviewer suspected that authors’ intention to detect “Agno RNA” was to cover all late gene massages, which are known to be alternatively spliced to generate multiple types of mRNA, some of which encode VP1. The method (or primer set) for detecting Agno RNA was not proven (for its effectiveness in previous studies). It needs to be shown in this manuscript that it holds sufficient sensitivity to detect the Agno coding messages by controlled experiments.
Lines 374 and 377, “retro-transcribed”, “retrotranscription”. It is more common to use “reverse-transcribed”, “reverse transcription”.
The addition of following data would enhance significance of this manuscript.
1.DNA sequence information of full length viral DNA in the positive samples.
2.Close examination of sequence analysis of the early gene, if it contains modification as found in MCPyV’s.
Author Response
Response to the Comments of Reviewer 2:
We thank this Reviewer for his/her comments. Our reply to the specific points is as follows:
Q1: Table 1 is highly confusing. The authors should devise way for better presentation. Alignment error of the numbers is unacceptable. Misplaced emphasis of data must be corrected. The significance of the table lies on the presence of JCPyV detected by DNA, RNA, and protein in each sample. Simply showing (+) or (-) attached after the numbers was not good way to attract readers’ attention. Creating categories, positive/ negative, may improve the clarity.
R.: Sorry for the inconvenience in formatting the table. We re-wrote the Table. The “positive”/ “negative” categories now substitute the plus and minus signs, to make the table clearer.
Q2: The reviewer suspected that authors’ intention to detect “Agno RNA” was to cover all late gene massages, which are known to be alternatively spliced to generate multiple types of mRNA, some of which encode VP1. The method (or primer set) for detecting Agno RNA was not proven (for its effectiveness in previous studies). It needs to be shown in this manuscript that it holds sufficient sensitivity to detect the Agno coding messages by controlled experiments.
R.: Our goal was to check if there was replicative activity of JCV in the colonic epithelial cells of the patients under study. As known, the late message of JCV is coded by a primary transcript, spliced from two introns located downstream the agno gene, or, as recently demonstrated (Saribas S. et al. Discovery and characterization of novel trans-spliced products of human polyoma JC virus late transcripts from PML patients. J Cell Physiol 2018 May;233(5):4137-4155), from alternatively trans-spliced sequences located downstream agno. We have amplified the agno transcripts with primers reported not previously, but, to our knowledge, any study reports the presence of alternative splicing of agno RNA sequence. Considering the reviewer's criticism regarding the sensitivity required to detect the agno mRNA, we believe that the absence of viral replication is supported by the lack of Vp1 transcription, whose effectiveness was proved in previous studies. To avoid problems, therefore, we eliminated the data related to the detection of agno mRNAs.
Q3: Lines 374 and 377, “retro-transcribed”, “retrotranscription”. It is more common to use “reverse-transcribed”, “reverse transcription”.
R.: OK, done.
Q4: The addition of following data would enhance significance of this manuscript.
1.: DNA sequence information of full length viral DNA in the positive samples.
R.: We agree completely with the Reviewer. We couldn't sequence the full length viral DNA. For some of our samples, we have rolling circle PCR data, which may suggest a possible integration of the virus, However, the data are not rigorous enough to deserve publication.
2.: Close examination of sequence analysis of the early gene, if it contains modification as found in MCPyV’s.
R.: We agree with the Reviewer. Unfortunately, time restrictions (and also sample restrictions) do not allow this examination, for the present paper.